# Abdominal ultrasound in the diagnostic work-up of visceral leishmaniasis and for detection of complications of spleen aspiration

Rezika Mohammed[1,2]*, Yonathan Gebrewold[3], Angela Schuster[4¤], Helina Fikre[2], Tigist Mekonnen[2], Tadele Mulaw[2], Tadfe Bogale[2], Florian Vogt[4], Ermias Diro[1], Johan van Griensven[4]

1 Department of Internal Medicine, University of Gondar, Gondar, Ethiopia, 2 Leishmanisis Research and Treatment Center, University of Gondar, Gondar, Ethiopia, 3 Department of Radiology, University of Gondar, Gondar, Ethiopia, 4 Department of Clinical Sciences, Institute of Tropical Medicine, Antwerp, Belgium

¤ Current address: Charité –Universitätsmedizin Berlin, corporate member of Freie Universität Berlin, Humboldt-Universität zu Berlin, and Berlin Institute of Health, Institute of General Practice, Berlin, Germany
* rezikamohammed@yahoo.com

**Data Availability Statement:** All relevant data are within the manuscript and its Supporting Information files.

## Abstract

### Introduction

Abdominal ultrasound (US) is increasingly used in the diagnostic work-up of infectious diseases, but studies on its diagnostic value in visceral leishmaniasis (VL) are lacking. US could help to identify complications of spleen aspiration (SA). We aimed to assess the diagnostic value of US and the evolution of findings after VL treatment; the incidence and degree of splenic injury; and the pain perceived during SA.

### Methodology/result

We conducted a cross-sectional prospective study at the Leishmaniasis Research and Treatment Center, Gondar, Ethiopia between Oct 2017 and Dec 2018. We enrolled VL suspects undergoing tissue aspiration; US were conducted before and after SA, and at the end of VL treatment. Splenic injury was graded using the American association of surgery trauma injury scale (grade 1–4). The pain perceived during SA was graded using a visual analogue scale. Out of 392 VL suspects, 192 (49%) were confirmed VL cases. The median age was 25 years (IQR 21–30). Massive splenomegaly and hepatomegaly were the most common US findings. Splenic nodules were seen in 3.7% of the 190 VL cases and 1.5% of the 197 non-VL cases. Ascites was more common in VL (16.4%) than in non-VL cases (9.1%). The frequency of US abnormalities decreased with treatment. None of the US findings had sufficient sensitivity and specificity to justify its use as a diagnostic test. US detected splenic injury in four of the 318 patients who had post-SA US. All four patients remained clinically stable. Pain was perceived as moderate or severe in 51% of patients.

**Funding:** The authors received no specific funding for this work.

**Competing interests:** The authors have declared that no competing interests exist.

## Conclusion

The diagnostic value of abdominal US for VL was low but found useful to detect subclinical splenic injury. SA caries a risk of splenic injury and was perceived painful by most. Further research on less invasive diagnostic tools is needed.

### Author summary

Diagnostic value of abdominal US for visceral leishmaniasis in Ethiopian setting is limited. US finding of patients decreased in frequency with VL treatment. US detects iatrogenic subclinical splenic injury trough splenic aspiration in VL suspects. Iatrogenic splenic injury occurred in 1.3%of all blinded splenic aspirations done in VL suspected patients. 51% of VL suspects described moderate to severe pain (score $\geq$ 5 on visual analogue scale) after splenic aspiration.

## Introduction

Visceral leishmaniasis (VL) is a parasitic disease caused by the *Leishmania donovani* complex. The parasite targets cells of the reticulo-endothelial system, predominantly bone marrow, lymph nodes, liver and spleen. Granulomatous lesions can develop in the affected organs [1]. Clinically, the disease presents with prolonged fever, hepatosplenomegaly, pancytopenia, weight loss and bleeding. Untreated, VL is usually fatal. Over 90% of global cases are reported from seven countries: Brazil, Ethiopia, India, Kenya, Somalia, Sudan and South-Sudan [2].

In countries such as Ethiopia, guidelines recommend treating VL based on clinical grounds and a positive rK39 rapid diagnostic test (RDT), a serological test. However, in East Africa, rK39 RDT sensitivity has been found to be suboptimal, particularly in HIV-coinfection, a common condition in North-West Ethiopia [3,4]. Additionally, serological tests do not help to diagnose VL relapse and cannot distinguish latent infection from active disease [3]. Consequently, the diagnosis of VL in this setting still depends on invasive procedures such as spleen aspiration (SA), while novel non-invasive diagnostic tools remain to be evaluated.

Ultrasound (US) has been used for a variety of infectious diseases, including tuberculosis and neglected tropical diseases such as schistosomiasis and echinococcosis [5,6]. Over the last decade, US is increasingly recommended as a diagnostic tool for infectious diseases and point-of-care US applications are particularly useful in resource-limited settings [6].

Only a few studies have reported on abdominal US findings in VL patients [6]. In several case reports, VL was characterized by focal [7–9] or granulomatous lesions in the spleen and/or liver [10] which could be visualised by US, CT scan or MRI. In a study in 2014 that included 50 Sudanese VL patients hepatosplenomegaly, ascites and intra-abdominal lymphadenopathy were reported as common abdominal US findings [11]. Another study, also from Sudan, described focal splenic lesions in 34% of 50 VL patients [12]. Multiple nodular hypo- or hyperechogenic lesions with or without halo have also been found previously in sporadic case reports [8,11,13,14]. However, none of these studies included a control group; hence the diagnostic accuracy of US for VL diagnosis remains undefined to date.

Given the insufficient accuracy of the rK39 RDT in countries such as Ethiopia, tissue aspiration from spleen or bone marrow is still commonly performed in routine care to detect the parasite. Additionally, tissue aspiration is performed as a test of cure in HIV-infected patients

or those with a slow treatment response, and systematically during clinical trials in VL patients. SA is the most common parasitological diagnostic method in East-Africa [15]. Besides being a painful procedure, splenic aspiration carries the risk of splenic laceration, rupture and potentially fatal bleeding. As to East-Africa, the evidence-base on its safety comes from a single-site report from Kenya, conducted over 35 years ago, in which no increased risk of major bleeding was found in over 600 SA done [16]. Since then, no systematic safety evaluation of SA in VL patients has been conducted in routine care settings in East-Africa. The recent identification of a severe splenic haemorrhage during the course of a clinical trial in Gondar suggests that this complication might not be so rare, and might potentially be missed [17]. There are also no studies assessing the severity of pain perceived during this procedure.

Amongst VL suspects in a tertiary VL treatment center in North-West Ethiopia, we aimed to assess 1) the diagnostic accuracy of abdominal US and the evolution of US findings after VL treatment, 2) the incidence and degree of (subclinical or clinical) splenic injury after SA, and 3) the severity of pain perceived by patients during the SA procedure.

## Methodology

### Ethics

Ethical approval was obtained from the University of Gondar Institutional Review Board, the Institutional Review Board of the Institute of tropical Medicine and the Ethics Committee of the University of Antwerp, Belgium. Written informed consent was obtained from all adult patients/ parents/guardians and an assent for patients aged 12 years and above included into the study. Patients received the standard of care as per the Ethiopian national guidelines for the treatment of VL and related medical conditions.

### Study setting and population

The study was conducted at the Leishmania Research and Treatment Center (LRTC) at the University of Gondar in North-West Ethiopia. The center, supported by the Drugs for Neglected Disease Initiative (DND*i*), conducts clinical research on VL and provides free health care to VL patients. The radiology department of the University of Gondar provides imaging services (including US) for all patients seen in the hospital, including those from LRTC.

All VL suspected patients with an indication for tissue aspiration, either coming directly to LRTC or being referred from local health centers or from other departments of the university hospital were screened for eligibility between October 2017 and December 2018. Patients with severe/unstable medical conditions or those not willing to have splenic or bone marrow aspiration or US were not included. Patients were also not included if presenting late in the afternoon and/or when no radiologist was available to conduct the US on the day of the procedure (splenic aspiration or bone marrow).

### Study design and sample size

A prospective cross-sectional study was conducted to assess the diagnostic accuracy of abdominal US for VL and to record the incidence of splenic injury and severity of pain perceived during SA. The sensitivity of US for diagnosing VL is unknown. Thus, with a conservative estimate of sensitivity of 50%, with a precision of 7.5%, considering the prevalence of VL to be 50% among the VL suspects, and with 10% refusals, a sample size of 394 was calculated. All eligible patients were consecutively enrolled.

## Study procedures

VL diagnosis and Treatment: At LRTC, all patients with typical VL symptoms (fever for two or more weeks, loss of appetite or weight loss and splenomegaly or lymphadenopathy) with a travel history to or residents of a VL endemic area were clinically considered as VL suspects. Except for patient with contraindications, in which bone marrow aspiration is preferred, SA was performed to confirm the diagnosis. Contraindications for doing spleen aspiration were signs of active bleeding, jaundice, critically ill patients, severe anaemia (hemoglobin count ≤ 5 g/dL) and platelet count < 40,000/ml. SA was done blindly (not US guided) with a 21 gauge needle attached to a 5 ml syringe to create vacuum and aspirate tissue by a quick in-out motion [18]. Giemsa-stained slides were microscopically examined for the presence of the leishmania parasite and positive slides graded from one to six depending on the concentration of the amastigotes [19]. At LRTC, non VL cases with no alternative diagnosis were sent to general medical outpatient or inpatient departments of the hospital for further investigation and decision. Based on clinician decision, some patients clinically suspected to have VL despite a negative parasitological test could have been treated for VL based on rK39 RDT results, but this was only exceptionally done. A full blood count was routinely done for all patients. Following SA, the vital signs were monitored every 30 minutes for the following four hours. As per national guidelines, all VL cases were tested for HIV and if positive received HIV care at the hospital antiretroviral treatment clinic.

First line VL treatment consisted of sodium stibogluconate combined with paromomycin for 17 days. Alternatives include liposomal amphotericin B and miltefosine.

Definitions: VL cases were defined as VL suspects that have leishmania parasites detected in the tissue aspirate by microscopy; non VL cases were those without leishmania parasites. Anaemia was defined as hemoglobin levels < 12 g/dL for females and < 13 g/dL for males. Severe anaemia was defined as hemoglobin < 5 g/dL. Thrombocytopenia was classified as severe (< 50x10³/μL), moderate (50-100x10³/μL) and mild (100-150x10³/μL) and leukopenia as a white blood cell count < 4000/μL.

Ultrasound: In this study, abdominal US was conducted before tissue aspiration and 4–6 hours after SA in order to detect complications such as splenic haemorrhage. For VL cases, US were repeated at the end of treatment to assess the evolution of US findings. The abdominal US procedure was done in line with the WHO guidelines [20]. US were done using a Sono-Scope machine, with patients lying in supine position. Scanning was performed using a 3.5 – 5mhz sector probe, with a high frequency linear probe used when necessary (*e.g.* to confirm miliary nodules in the spleen). Color-doppler scanning was employed to differentiate between nodules and blood vessels. The abdomen was scanned in the following sequence: liver, pancreas, spleen, kidney, peritoneum and mesentery and other retroperitoneal organs. The American association of surgery trauma injury scale was used to grade splenic injury from one to four and type of injury as laceration or hematoma [21]. The spleen size was measured from pole to pole. Splenomegaly was defined as a size of 13 cm and above and as massive splenomegaly when the size was more than 20 cm. Hepatomegaly was defined when the longitudinal hepatic right lobe size was more than 15 cm. Lymphadenopathy was defined as lymph nodes > 1cm in short axis. Spleen, liver and kidney echotexture was defined as normal when the tissue had a homogenous echogenicity. Any change from the normally expected echogenicity was interpreted as abnormal.

Assessment of pain associated with SA: The visual analogue scale (VAS) was used to assess pain associated with SA immediately after the procedure. The VAS score varies between 0 (no pain) and 10 (severe pain). A score of 1–4 was considered as mild pain; 5–6 as moderate and 7–10 as severe pain.

### Data collection and analysis

Data was collected by trained nurses and physicians using a paper-based case report form. The patient medical files were also used as a source of data. The study followed Standards for the Reporting of Diagnostic Accuracy Studies (STARD) guidelines (S1 Table). Four radiologists were involved in the study to do the US. Three of them were at the end of their second year radiology residency course and had done more than 500 abdominal US. One senior radiologist was regularly doing quality checks of the US procedure. The radiologists performing the US were blinded to the clinical information and VL test result. Data were entered into Epi Info software version 7.0.8.3. Frequency (%), medians (interquartile range (IQR)), sensitivity, specificity and the likelihood ratio (LHR) of the different US findings was calculated using Microsoft XL version 16.0 dated Nov 2016. Comparisons between groups were based on the Wilcoxon rank-sum test for continuous variables and the Fisher's exact test for binary/categorical data. Within patient comparisons were based on the Mc Nemar test. Patients with missing data on the US findings were not included in the US analysis. Raw data are provided (S1 Raw data set).

## Results

A total of 1583 patients were referred to LRTC between October 2017 and December 2018 based on signs and symptoms or rK39 RDT test results. Of these, 514 underwent tissue aspiration. A total of 392 VL suspects were included in the study and VL was parasitologically confirmed in 192 (49%) (Fig 1). The study population was predominantly male (n = 369; 98%) with a median age of 25 years (IQR 21–30). The majority of the VL suspects (302/392; 76.8%) presented to the hospital within three months of symptom onset. SA was done for 342 (87.3%) the remaining patients underwent bone marrow aspiration (n = 50; (12.7%)).

VL cases were younger, had a lower body mass index and were more likely to have hepatomegaly. Splenomegaly was clinically present in almost all patients, but less likely to be massive in VL cases. The proportion of other infections such as HIV, tuberculosis or malaria was

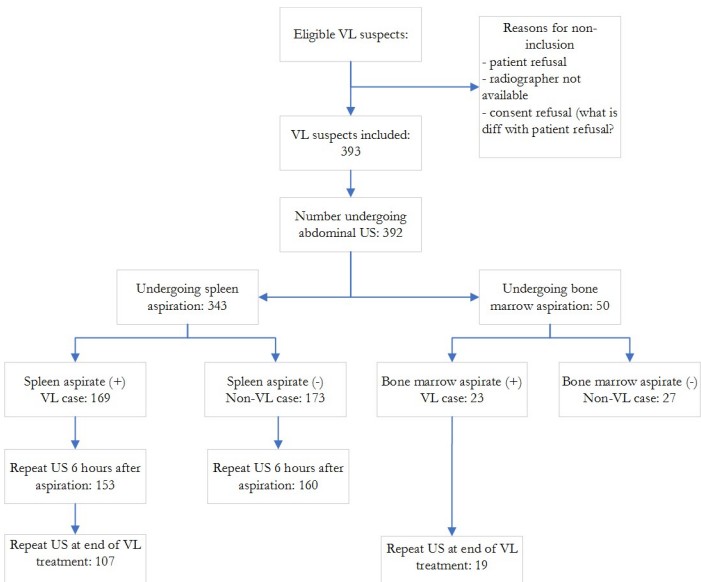

**Fig 1. Overview of patients enrolled in the study and procedures performed.** US: ultrasound; VL: visceral leishmaniasis.

17.6% in VL and 22.5% in non VL cases (Table 1). Leukopenia and thrombocytopenia was more common in VL cases compared to non VL cases but the prevalence of anaemia was similar in both groups (Table 2).

**Table 1. Socio-demographic and clinical characteristics of VL suspected patients undergoing tissue aspiration at the Leishmaniasis Research and Treatment Center, Gondar Ethiopia (Oct 2017-Dec 2018; N = 392).**

| Baseline characteristics | VL (n = 192) | Non VL (n = 200) | P value |
|---|---|---|---|
| Age (years), median (IQR); (n = 392) | 23.5 (20.0–28.0) | 27.0 (22.0–33.0) | <0.001 |
| < 18 | 5 (2.6) | 2 (1.0) | 0.001 |
| 18–24 | 98 (51.0) | 72 (36) | |
| 25–35 | 69 (35.9) | 80 (40.0) | |
| ≥ 35 | 20 (10.4) | 46 (23.0) | |
| Male sex, n (%); (n = 375) | 184 (99.5) | 185 (92.0) | 0.122 |
| History of VL, n (%); (n = 392) | 27 (14.0) | 33 (17.0) | 0.575 |
| Ability to walk, n (%); (n = 392) | 186 (97.0) | 197 (99.0) | 0.329 |
| Duration of symptoms (months), median (IQR); (n = 375) | 1.0 (0.8–2.0) | 1.0 (0.8–2.0) | 0.720 |
| < 1, n (%) | 53 (28.0) | 63 (34.0) | 0.016 |
| 1–3, n (%) | 101 (54.0) | 85 (45.0) | |
| 3–6, n (%) | 33 (18.0) | 29 (16.0) | |
| > 6, n (%) | 1 (1.0) | 10 (5.0) | |
| Jaundice, n (%); (n = 390) | 23 (12.0) | 28 (14.0) | 0.653 |
| Vomiting, n (%); (n = 388) | 44 (23.0) | 44 (22.0) | 0.904 |
| Lymphadenopathy, n (%); (n = 369) | 12 (7.0) | 17 (9.0) | 0.562 |
| Bleeding, n (%); (n = 382) | 58 (31.0) | 52 (27.0) | 0.431 |
| Body mass index (kg/m$^2$), median (IQR); (n = 384) | 16.9 (15.6–17.8) | 17.6 (16.7–18.7) | <0.001 |
| Normal (18.5–25), n (%) | 30 (16.0) | 58 (23.0) | < 0.001 |
| Mild malnutrition (16–18.5), n (%) | 97(52.0) | 113 (57.0) | |
| Moderate malnutrition (14–16), n (%) | 52 (28.0) | 24 (12.0) | |
| Severe malnutrition (< 14), n (%) | 8 (4.0) | 2 (1.0) | |
| Spleen size (cm), median (IQR); (n = 391) | 8.0 (5.3–11.0) | 10.0 (8.0–14.0) | < 0.001 |
| Splenomegaly;[a] n (%) | 187 (96.4) | 196 (98.5) | 0.005 |
| Mild (< 5 cm) | 29 (16.0) | 17 (9.0) | |
| Moderate (5–8 cm) | 41 (22.0) | 25 (13.0) | |
| Massive (> 8 cm) | 117 (63.0) | 154 (79.0) | |
| Ascites, n (%); (n = 381) | 16 (9.0) | 8 (4.0) | 0.091 |
| Hepatomegaly, [b] n (%); (n = 381) | 88 (47.0) | 71 (37.0) | 0.049 |
| Peripheral edema, n (%); (n = 355) | 39 (23.0) | 21 (12.0) | 0.007 |
| Presence of concomitant infection, n (%); (n = 337) | 24 (17.6) | 32 (22.5) | 0.646 |
| Tuberculosis | 3 (13) | 3 (9.0) | |
| Malaria | 1 (4.0) | 5 (16.0) | |
| Diarrhea | 6 (25.0) | 6 (19.0) | |
| Pneumonia | 12 (50.0) | 8 (25.0) | |
| Other | 2 (8.0) | 10 (31.0) | |
| HIV test, n (%); (n = 242) | | | 0.435 |
| Negative | 107 (80.0) | 83 (76.0) | |
| Positive | 26 (20.0) | 26 (24.0) | |

IQR: interquartile range; VL: visceral leishmaniasis

a Splenomegaly is clinically defined as palpable and measured in cm below the left costal margin

b Hepatomegaly clinically defined as palpable and measured in cm below the right costal margin

**Table 2. Laboratory profile of VL suspected patients undergoing tissue aspiration at the Leishmaniasis Research and Treatment Center, Gondar Ethiopia (Oct 2017-Dec 2018; N = 392).**

| | VL (n = 192) | Non-VL (n = 200) | P value |
|---|---|---|---|
| Blood platelets, median (IQR); (n = 388) | 64000 (43000–130000) | 81000 (55000–122000) | <0.001 |
| Thrombocytopenia[a], n (%); | 174 (92.0) | 199 (84.7) | 0.006 |
| Hemoglobin (g/dL), median (IQR); (n = 390) | 9.1 (7.7–10.2) | 9.2 (7.1–10.9) | 0.768 |
| Anaemia[b], n (%) | 174 (91.0) | 171 (86.0) | 0.116 |
| White blood cell count/μL, median (IQR); (n = 379) | 1700 (1400–2450) | 3500 (23750–4600) | <0.001 |
| Leukopenia[c], n (%) | 176 (95.0) | 119 (61.0) | <0.001 |
| rK 39 RDT, n (%); (n = 345) | | | <0.001 |
| Negative | 1 (1.0) | 57 (33.0) | |
| Positive | 167 (99.0) | 118 (67.7) | |
| Tissue aspiration site, n (%); (n = 392) | | | 0.762 |
| Spleen aspirate | 169 (88.0) | 173 (87.0) | |
| Bone marrow | 23 (12.0) | 27 (14.0) | |

RDT: rapid diagnostic test

[a] Defined as platelet count $< 150 \times 10^3/\mu L$

[b] Defined as hemoglobin levels $< 12$ g/dL for females and $< 13$ g/dL for males

[c] Defined as a white blood cell count $< 4000/\mu L$

The most common US abnormality was splenomegaly, seen in all but two VL suspects. Splenomegaly was massive (>20 cm) in 97 (51.3%) and 108 (55.7%) of VL and non VL cases respectively (Table 3). Splenic nodules were seen in 7 (3.7%) of the 190 VL cases and 3 (1.5%) of the 197 non-VL cases. In both groups splenic nodules were hyperechoic, small, multiple and non-caseating. Hepatomegaly was also common, especially in VL cases. Focal lesions in the liver were seen in one VL case and six non-VL cases. Intra-abdominal lymphadenopathies were seen in two VL cases and three non-VL cases patients. Echotexture of the organs was mostly normal. Whereas ascites was more common in VL (31/189; 16.4%) than in non VL patients (18/197; (9.1%)), the reverse was true for pleural effusion. None of the US findings were sensitive nor specific. Most of the LHR of the US findings were < 2 or > 0.5, hence yielding limited confirming or excluding power. The few abnormalities scoring > 2 or < 0.5 occurred in less than 5% of the suspects.

Of the five patients with abdominal lymphadenopathies on US, one was diagnosed with smear-negative tuberculosis supported by the US findings, one VL patient died before further investigations could be done, one VL patient had no other concurrent diagnosis. Two non VL patients had HIV with no other diagnosis made. From 10 patients with splenic nodules (seven VL and 3 non VL), none was diagnosed to have tuberculosis or other relevant diseases explaining the splenic nodules. US findings were similar for patients with or without HIV (S2 Table).

For 126 (65.6%) of the 192 VL cases, a follow-up US was done at the completion of treatment, with most US abnormalities decreasing in frequency or severity. The median duration between the initial and follow-up US was 17 days (IQR, 17–28). Splenomegaly was still present in all but one. Massive splenomegaly was only seen in 13.5% of the VL cases, as compared to 51.3% before treatment. All but one splenic nodule disappeared. Hepatomegaly was still present in 79 (65.0%). Abnormal echotexture of spleen and liver, ascites and pleural effusion were less frequent after treatment (Table 4).

Out of the 318 patients undergoing SA and with a post-aspiration US conducted, 4 (1.3%) cases of splenic injury were detected. On a severity scale of 1–4, three were grade one, one was grade two (Table 5). All four patients remained clinically stable during observation. One of

**Table 3. Sensitivity and specificity of abdominal ultrasound finding in VL suspects at the Leishmaniasis Research and Treatment Center, Gondar, Ethiopia (Oct 2017 –Dec 2018; N = 392)***.

| Ultrasound finding; n (%) | VL; n (%) | Non VL; n (%) | Likeli-hood ratio | Sensitivity (%) | Specificity (%) |
|---|---|---|---|---|---|
| **Ascites** | | | | | |
| No | 158 (84.0) | 179 (91.0) | 0.9 | | |
| Yes | 31 (16.0) | 18 (9.0) | 1.8 | 16.4 | 90.9 |
| Grade 1[γ] | 14 (7.4) | 10 (5.1) | 1.5 | | |
| Grade 2 | 8 (4.2) | 3 (1.5) | 2.8 | | |
| Grade 3 or 4 | 2 (1.1) | 0 (0.0) | - | | |
| **Pleural effusion** | | | | | |
| No | 188 (99.0) | 187 (95.0) | 1.0 | | |
| Yes | 2 (1.0) | 9 (5.0) | 0.2 | 1.1 | 95.4 |
| **Splenic echotexture** | | | | | |
| Normal | 184 (97.0) | 193 (97.0) | 1.0 | | |
| Abnormal | 5 (3.0) | 5 (3.0) | 1.0 | 2.6 | 97.5 |
| **Spleen enlargement[a]** | | | | | |
| No | 1 (1.0) | 3 (0.5) | | | |
| Yes | 188 (99.5) | 194 (99.5) | 1.0 | 99.5 | 0.5 |
| Mild/moderate | 91 (48.1) | 86 (44.3) | 1.1 | | |
| Massive | 97 (51.3) | 108 (55.7) | 0.9 | | |
| **Splenic nodules** | | | | | |
| No | 183 (96.0) | 194 (98.0) | 1.0 | | |
| Yes | 7 (4.0) | 3 (2.0) | 0.7 | 3.7 | 98.5 |
| **Lymphadenopathy** | | | | | |
| No | 188 (99.0) | 195 (98.0) | 1.0 | | |
| Yes | 2 (1.0) | 3 (2.0) | 0.7 | 1.1 | 98.5 |
| **Hepatomegaly[b]** | | | | | |
| No | 44 (24.0) | 76 (39.0) | 0.6 | | |
| Yes | 143 (76.0) | 119 (61.0) | 1.2 | 76.5 | 39.0 |
| **Liver echotexture** | | | | | |
| Normal | 181 (95.0) | 185 (93.0) | 1.0 | | |
| Abnormal | 9 (5.0) | 13 (7.0) | 0.7 | 4.7 | 93.4 |
| **Liver focal lesion** | | | | | |
| No | 189 (99.0) | 193 (97.0) | 1.0 | | |
| Yes | 1 (1.0) | 5 (3.0) | 0.2 | 0.5 | 97.0 |
| **Bile duct** | | | | | |
| Normal | 188 (100) | 195 (99.0) | 1.0 | | |
| Abnormal | 0 (0) | 2 (2.0) | - | 0 | 98.5 |
| **Pancreatic focal lesion** | | | | | |
| No | 188 (100) | 195 (100) | 1.0 | | |
| Yes | 0 (0) | 0 (0) | - | 0 | 100.0 |
| **Renal echotexture** | | | | | |
| Normal | 185 (98.0) | 192 (97.0) | 1 | | |
| Abnormal | 4 (2.0) | 5 (3.0) | 0.5 | 1.6 | 97.0 |
| **Renal focal lesion** | | | | | |
| No | 188 (99.0) | 196 (99.0) | 1.0 | | |

(*Continued*)

**Table 3.** (Continued)

| Ultrasound finding; n (%) | VL; n (%) | Non VL; n (%) | Likeli-hood ratio | Sensitivity (%) | Specificity (%) |
|---|---|---|---|---|---|
| Yes | 1 (1.0) | 2 (1.0) | 0.5 | 0.5 | 99.0 |

VL: visceral leishmaniasis; NA: not available

a Normal spleen size measured from pole to pole, normal spleen size < 12 cm, splenomegaly 13–20 cm and massive splenomegaly > 20cm.

b Hepatomegaly defined as longitudinal hepatic right lobe size ≥ 15cm

* Some US findings missing for some patients

γ Grade missing for some patients

these patients had VL and was discharged cured after VL treatment. At the time of discharge, the hematoma had disappeared on US. Two of the non VL cases were newly diagnosed with HIV and linked to the ART clinic. Concurrently, hyperactive malaria splenomegaly was diagnosed and anti-malarial treatment started. The fourth patient (a non VL case) was admitted for further investigation of the cause of the fever but discharged after a week with spontaneous improvement without conclusive diagnosis.

The pain score using a visual assessment scale (VAS) was assessed for 329 (95.9%) out of 342 patients undergoing SA. Pain intensity was comparable for VL and non-VL cases, with most (69.6%) perceiving the pain as mild (score 1–4) and 25 (7.6%) out of 329 perceiving the pain as severe (score 7–10) (Table 6).

After the SA, all patients remained clinically stable. An increase in pulse rate of > 25% from baseline was seen in 36 patients, but all rates were < 125 beats/minute. No clear increases in respiratory rates were seen.

## Discussion

The importance of abdominal US in diagnosing VL and assessing complications of splenic aspiration was evaluated in this study. In terms of diagnosing VL, US helped to detect some nodules, lymph nodes and ascites that might not have been detected clinically in addition to ascertaining organomegalies. However, these findings were detected in few of the cases and/or lacked confirmatory power (low LHR).

Most US abnormalities occurred with similar frequency in VL and non-VL cases, and splenic nodules were rare. The frequent detection of ascites and pleural effusion in our study is in line with findings from Sudan, as well as the detection of abnormal echotexture in kidneys, liver and spleen. Splenic focal lesions in VL were commonly reported from Sudan, occurring in 34% [11]. These were uncommon in VL patients in our study, and the frequency was not different in non-VL patients. Similarly, abdominal lymphadenopathy was infrequent in our study, but present in 35–90% of the VL cases in Sudan [11]. The reason for these differences is currently unclear. We note that, compared to VL in Ethiopia, peripheral lymphadenopathy is also reported to be more common in Sudan [22].

Overall, our findings argue against the routine use of abdominal US in the diagnostic work-up for VL, in this region of Ethiopia, where other granulomatous and infectious conditions are common. Most of the LHRs of the US findings had limited confirming or excluding power. The few abnormalities with LHRs of > 2 or < 0.5 occurred in less than 5% of the suspects hence limiting their clinical usefulness (Table 3). Consequently, routine US would not substantially alter the diagnostic probability of VL.

Current VL diagnostic tests neither ruled out VL nor could diagnose other conditions in the non-VL patients. US could potentially detect other diseases in such patients. Of the five patients with abdominal lymphadenopathies on US, one was diagnosed with smear-negative

**Table 4. Abdominal ultrasound findings of VL patients before and after treatment at the Leishmaniasis Research and Treatment Center, Gondar, Ethiopia (Oct 2017 –Dec 2018; N = 126)[*].**

|  | Before VL treatment; n (%) | After VL treatment; n (%) | P Value |
|---|---|---|---|
| **Ascites; n (%)** |  |  | 0.0784 |
| No | 105 (84.1) | 114 (91.3) |  |
| Yes | 19 (15.1) | 10 (7.9) |  |
| Grade 1[γ] | 11 (8.7) | 3 (2.4) |  |
| Grade 2 | 4 (3.2) | 2 (1.6) |  |
| Grade 3 or 4 | 0 (0.0) | 1 (0.8) |  |
| **Pleural effusion; n (%)** |  |  | 1.0 |
| No | 124 (97.6) | 125 (98.4) |  |
| Yes | 2 (1.6) | 1 (0.8) |  |
| **Splenic echotexture; n (%)** |  |  | 1.0 |
| Normal | 121 (96.4) | 123 (97.6) |  |
| Abnormal | 4 (3.2) | 3 (2.4) |  |
| **Splenomegaly[a]; n (%)** |  |  |  |
| Yes | 124 (98.4) | 123 (97.6) |  |
| Mild/moderate | 68 (54.0) | 103 (84.1) |  |
| Massive | 56 (44.4) | 17 (13.5) |  |
| **Splenic nodule; n (%)** |  |  | 0.500 |
| No | 123 (97.6) | 125 (99.2) |  |
| Yes | 3 (2.4) | 1 (0.8) |  |
| **Lymphadenopathy; n (%)** |  |  | 0.219 |
| No | 125 (99.2) | 121 (96.0) |  |
| Yes | 1 (0.8) | 5 (4.0) |  |
| **Hepatomegaly[b]; n (%)** |  |  | 0.043 |
| No | 31 (25.00) | 43 (35.0) |  |
| Yes | 91 (75.0) | 79 (65.0) |  |
| **Liver echotexture; n (%)** |  |  | 0.625 |
| Normal | 122 (96.8) | 124 (99.2) |  |
| Abnormal | 4 (3.2) | 2 (1.6) |  |
| **Liver focal lesion; n (%)** |  |  | 1.0 |
| No | 125 (99.2) | 124 (98.4) |  |
| Yes | 1 (0.8) | 2 (1.6) |  |
| **Bile duct; n (%)** |  |  | NA |
| Normal | 124 (97.6) | 126 (100.0) |  |
| Abnormal | 0 (0.0) | 0 (0.0) |  |
| **Pancreatic focal lesion; n (%)** |  |  | NA |
| No | 126 (100.0) | 126 (100.0) |  |
| Yes | 0 (0.0) | 0 (0.0) |  |
| **Renal echotexture; n (%)** |  |  | 0.125 |
| Normal | 123 (97.6) | 118 (93.7) |  |
| Abnormal | 2 (1.6) | 7 (5.6) |  |
| **Renal focal lesion; n (%)** |  |  | 1.0 |
| No | 124 (99.0) | 124 (99.0)) |  |
| Yes | 1 (1.0) | 1 (1.0) |  |

VL: visceral leishmaniasis; NA: not available

a Normal spleen size measured from pole to pole, normal spleen size < 12 cm, splenomegaly 13–20 cm and massive splenomegaly > 20cm.

b Hepatomegaly, longitudinal hepatic right lobe size ≥ 15cm

[*] Some US findings missing for some patients

γ Grade missing for some patients

**Table 5. Details of VL suspect patients with splenic haemorrhage after undergoing splenic aspiration at the Leishmaniasis Research and Treatment Center, Gondar, Ethiopia (Oct 2017 –Dec 2018).**

| Nr | VL status | Age | US grade | Description of splenic haemorrhage | Monitoring post SA | Pain score | Outcome |
|---|---|---|---|---|---|---|---|
| 1 | VL | 28 | I | Hematoma, subcapsular collection lower margins of the spleen; 7.8 mm in depth and capsular tear | Yes, clinically stable | 2 | Disappeared on repeat US; discharged alive |
| 2 | No VL | 24 | I | No details | Yes, clinically stable | 2 | Discharged alive; diagnosed as HMS; linked to HIV care |
| 3 | No VL | 35 | II | Laceration, linear hypoechoic lesion; 3 cm | Yes, clinically stable | 2 | Hospitalized for fever diagnostic work-up for one week; Discharged alive |
| 4 | No VL | 28 | I | Hematoma, 5.2 x2.8 cm crescentic shaped | Yes, clinically stable | 5 | Discharged alive; diagnosed as HMS; linked to HIV care |

HSM: hyper-reactive malaria splenomegaly; US: ultrasound SA: spleen aspiration; VL: visceral leishmaniasis

tuberculosis supported by the US findings. We however acknowledge that some cases of tuberculosis could have been missed. Nevertheless, with no other relevant non-VL diagnoses made, the overall diagnostic value of US in our setting was limited.

While our findings could be considered 'negative', these nevertheless contribute to the evidence. Previous studies were relatively small case-series and none included non VL cases as controls. Our study questions the value of abdominal US in VL diagnosis as part of routine care in VL suspect patients. It remains to be further assessed whether the procedure would be useful in HIV patients with suspected VL, as concurrent tuberculosis has been reported to be very common in this group and US could contribute to the diagnosis of disseminated or abdominal tuberculosis [23]. Comparative studies in Ethiopia and Sudan would also shed light on the discrepancies in findings seen in our and reported studies from Sudan.

The frequency of massive splenomegaly decreased after treatment but only two patients had a normal spleen size at the end of treatment. While normalisation of spleen size after VL treatment might require more time, future studies with longer follow-up should also assess whether conditions such as schistosomiasis and malaria, which are common in the study areas, contribute to the persistence of splenomegaly in this group of patients.

Very importantly, US helped to detect splenic injury after SA. Four (1.3%) of the patients who underwent SA had splenic injury manifested as laceration or hematoma. Although the final outcomes of the patients were good, the findings are important to alert physicians to take extra precautions during SA as well post procedure care. The prevalence of splenic injury from SA is not rare as was reported in a few studies before. Post SA uncontrollable bleeding from the spleen that led to death has been noticed in our setup before [17].

**Table 6. Visual analogue pain score associated with splenic aspiration in VL suspects at the Leishmaniasis Research and Treatment Center, Gondar, Ethiopia (Oct 2017 –Dec 2018: N = 329).**

| Pain Score | Total; n (%) |
|---|---|
| 0 (no pain) | 13 (3.9) |
| 1–2 | 5 (1.5) |
| 3–4 | 144 (43.8) |
| 5–6 | 124 (37.7) |
| 7–8 | 27 (8.2) |
| 9–10 | 16 (4.9) |
| Total | 329 |

VL: visceral leishmaniasis

In high-income countries, SA–usually conducted under US guidance—is considered to be safe. While in resource-constrained setting, this procedure is routinely done without US guidance, it is reportedly to be safe as well despite an overall weak evidence-base. Though clinical implications of splenic injury remained limited, this suggests a non-negligible risk, especially since our study was conducted in a tertiary hospital with highly qualified and experience staff. While large studies would allow more reliable estimates of its' occurrence and the clinical significance, our findings underscore that close observation after SA remains warranted and that a careful risk-benefit balance remains merited. The observation that more than 50% reported a VAS pain score of five or more should also be integrated in the decision-making process

There are a number of important limitations to our study. First, US was not conducted for some patients after SA. A repeated US for all and longer clinical follow-up in the patients with splenic injury would have allowed to more strongly define these as subclinical. Second, despite the use of standard operating procedures for the abdominal US, and the fact that radiologist were blinded on the VL status, US was performed by several radiologists, which might have induced interobserver bias particularly since US is an examiner and training dependent method. Third, VL patients with negative parasitological test may be misclassified as non-VL cases. Fourth, presence of coinfection was not investigated extensively at the time of study inclusion and final diagnosis for non VL cases were not known as patients were sent to other departments of the hospital. Fifth, small nodules might be missed as linear probe was only used to differentiate nodule from vessels not for screening. Finally, as the study was conducted in a center with good clinical expertise with tissue aspiration, complications might not be representative of the district settings. The strengths of this study include the prospective nature of the study. In contrast to the case reports conducted previously, we performed a formal diagnostic study, including non-VL cases. Pain during SA has not been assessed before.

In conclusion, the value of abdominal US to diagnose VL in Ethiopia was limited in our study. Subclinical splenic injury was detected in 4 out of 318 patients undergoing SA and the procedure was perceived as painful by most patients. The occurrence of splenic injury after SA emphasizes the need of close observation after the procedure and a careful risk-benefit assessment. If available, US post-aspiration could further add to the safety of the procedure for patients undergoing SA for detection of splenic injuries.

## Supporting information

**S1 Table. Ultrasound diagnosis of visceral leishmaniasis study STARD checklist filled.** (DOCX)

**S2 Table. Abdominal ultrasound finding in VL suspects at the Leishmaniasis Research and Treatment Center, Gondar, Ethiopia.** (DOCX)

**S1 Raw data set. Raw data set for the ultrasound study in VL suspects at Leishmaniasis research and Treatment center, Gondar, Ethiopia.** (XLSX)

## Acknowledgments

We thank all LRTC staff members who are actively involved in the management of these patients. Our acknowledgments also goes to the Drugs for Neglected Disease Initiative (DNDi) and University of Gondar (UoG) for supporting the Leishmaniasis Research and Treatment Center.

## Author Contributions

**Conceptualization:** Rezika Mohammed, Angela Schuster, Florian Vogt, Ermias Diro, Johan van Griensven.

**Data curation:** Rezika Mohammed, Angela Schuster, Helina Fikre, Tigist Mekonnen, Tadele Mulaw, Tadfe Bogale, Johan van Griensven.

**Formal analysis:** Rezika Mohammed, Florian Vogt, Johan van Griensven.

**Investigation:** Rezika Mohammed, Yonathan Gebrewold, Helina Fikre, Tigist Mekonnen, Tadele Mulaw, Tadfe Bogale.

**Methodology:** Rezika Mohammed, Helina Fikre, Tigist Mekonnen, Tadele Mulaw, Tadfe Bogale, Johan van Griensven.

**Resources:** Rezika Mohammed, Yonathan Gebrewold, Helina Fikre.

**Supervision:** Angela Schuster, Ermias Diro, Johan van Griensven.

**Validation:** Angela Schuster, Florian Vogt.

**Writing – original draft:** Rezika Mohammed.

**Writing – review & editing:** Rezika Mohammed, Yonathan Gebrewold, Angela Schuster, Helina Fikre, Tigist Mekonnen, Tadele Mulaw, Tadfe Bogale, Florian Vogt, Ermias Diro, Johan van Griensven.

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
