## [Decision Letter · Decision Letter 0]

27 Jul 2020

Dear Dr. Mohammed,

Thank you very much for submitting your manuscript "Abdominal ultrasound in the diagnostic work-up of visceral leishmaniasis and for detection of complications of spleen aspiration" for consideration at PLOS Neglected Tropical Diseases. As with all papers reviewed by the journal, your manuscript was reviewed by members of the editorial board and by several independent reviewers. In light of the reviews (below this email), we would like to invite the resubmission of a significantly-revised version that takes into account the reviewers' comments. 

We cannot make any decision about publication until we have seen the revised manuscript and your response to the reviewers' comments. Your revised manuscript is also likely to be sent to reviewers for further evaluation.

Sincerely,

Jaap J van Hellemond

Deputy Editor

Francesca Tamarozzi

Deputy Editor

Reviewer's Responses to Questions

**Key Review Criteria Required for Acceptance?**

**Methods**

-Are the objectives of the study clearly articulated with a clear testable hypothesis stated?

-Is the study design appropriate to address the stated objectives?

-Is the population clearly described and appropriate for the hypothesis being tested?

-Is the sample size sufficient to ensure adequate power to address the hypothesis being tested?

-Were correct statistical analysis used to support conclusions?

-Are there concerns about ethical or regulatory requirements being met?

Reviewer #1: I think this is a prospective study, which was very well conceived, executed and described. I would recommend accepting it with minimal revisions.

Reviewer #2: (No Response)

Reviewer #3: - I suggest to assess the differences in proportion of the variables in the VL and non-VL patients with statistical tests. The authors should report the p-values in table 1, table 2 and table 4

**Results**

-Does the analysis presented match the analysis plan?

-Are the results clearly and completely presented?

-Are the figures (Tables, Images) of sufficient quality for clarity?

Reviewer #1: The results are clearly stated.

Reviewer #2: (No Response)

Reviewer #3: (No Response)

**Conclusions**

-Are the conclusions supported by the data presented?

-Are the limitations of analysis clearly described?

-Do the authors discuss how these data can be helpful to advance our understanding of the topic under study?

-Is public health relevance addressed?

Reviewer #1: (No Response)

Reviewer #2: (No Response)

Reviewer #3: (No Response)

**Editorial and Data Presentation Modifications?**

Reviewer #1: Comments:

What was not clear to me is on which criteria the authors decided whether to do a SA or a bone marrow aspirate. Maybe this could be better clarified in the methods section.

Minor comments:

L34: remove bracket

L120 convenience (?)  121 sampling method  

L130 Patients who could  

L123-136 should be written in past tense as the rest of the section

L180 what was the descision criteria to do BMA vs SA?

L186 was more common

L 323 I would not say US is highly subjective, I think that is an anachronism and true for almost every medical examination – from listening to the sound of a heart to looking at a CXR. I would suggest “examiner- and training-dependent”.

L334 The last recommendation to preform post intervention US sounds strange in light of the fact that injuries were rare and none of the detected injuries led to consequences. I still understand (and agree) that it should be done- but maybe the statement could be tuned down. 

Table 1 spleen size median (IQR) vs Yes (cm below costal margin) 

Table 2 Anemia – is this in % or absolutes?

rK39 data is confusing – what are absolutes what are percent and what are the percentages on: I would suggest to use the percentage of those tested instead of percentage of all patients. I think table 2 needs revision regarding the units, %, IQR…

Reviewer #2: (No Response)

Reviewer #3: I suggest this article could be improved by some revisions: 

- There are some typing errors (capital letters, punctuation…);

- In the “Study procedures” section I suggest to indicate which kind of needle has been used for the spleen aspiration (in particular the gauge of the needle). I suggest to specify if US-guided spleen aspiration has been performed; 

- In the “data collection and analysis” section the authors mentioned that “radiologists” performed all the US. How many radiologists? How much experience have they in abdominal US? Interpretation and variability of the results could be influenced by these specifications;

- I have noticed that in this population there was a high proportion of HIV-positive patients. The most part of the US abnormalities have been evaluated (particularly splenic nodules and abdominal lymphadenopathies) are frequently detected also in HIV-patients with abdominal TB. In such a setting with a high prevalence of HIV/TB co-infection as Ethiopia, it would be possible to deal with abdominal TB very often. How did the authors definitively exclude abdominal TB in patients with these US abnormalities? There was any VL-patient with splenic nodules or abdominal lymphadenopathies who had a TB co-infection? I suggest to comment these aspects in the discussion;

- I suggest to describe US features of splenic nodules and hepatic lesions (micro-/macro-nodules, hypo/hyper-echoic, single/multiple/miliary…);

- I suggest to assess the differences in proportion of the variables in the VL and non-VL patients with statistical tests. The authors should report the p-values in table 1, table 2 and table 4;

- In table 5, point 3: I suppose the authors would mean “hypo-echoic lesion”;

- Line 219: I suppose the authors would refer to longitudinal hepatic right lobe diameter ≥ 15 cm.

**Summary and General Comments**

Reviewer #1: This is a first systematic study on the use of ultrasound in VL in an endemic setting. I is therefore novel and scientifically significant.

Reviewer #2: The authors present ultrasound and clinical data from a prospective study in patients with suspected visceral leishmaniasis (VL) in Ethiopia. The study cohort comprises around 400 patients with half of them having confirmed VL. Abdominal ultrasound, performed by the radiology department, was evaluated as diagnostic and follow-up test for VL and as a tool to detect splenic injury after diagnostic spleen aspiration. Additionally, the study evaluated severity of pain associated with spleen aspiration. The study addresses an important topic with clinical relevance and is generally well presented. However, there is considerable potential for improvement of the manuscript by elaborating more on the case definitions and co-infections (see comments below).

My main concerns with the data are the case definitions:

First, in non VL patients RDT was positive in 60% and diagnoses other than VL are reported for 23% only. How sure are you having excluded VL?

Second, 18% of VL patients had concomitant infections of which some are known to present with similar sonographic features as VL. As detailed data for these concomitantly infected patients are not presented separately the reader does not know if e.g. splenic nodules occurred in VL/TB co-infected patients and hence etiology of splenic nodules (TB or VL) remains unclear. Presenting/describing sonographic data for co-infected patients (or exclusion of co-infected patients) appears very important not only for a more accurate calculation of the diagnostic value but also for interpretation of follow-up data. Follow-up US after VL treatment did not lead to a resolution of all findings, there was even occurrence of some new findings. Did co-infected patients receive treatment for their co-infections in parallel?

If I understand correctly, patients were not systematically investigated for co-infections like TB (especially as 18% were HIV-infected and VL has been discussed as additional risk factor for TB). Therefore, even if you exclude co-infected patients from your cohort there remains the limitation of possible undetected co-infection. This should be highlighted in the limitations/discussion.

Given the considerable ratio of HIV infected patients in the VL (and non-VL group) I would be interested if you see any difference in US findings in HIV-infected versus non-infected (in TB HIV-co-infection has a significant impact on US findings).

Please find further comments below:

Abstract:

1. The abstract should include the study site and study period.

Introduction:

2. As US was also evaluated for follow-up, consider adding a short mention of the need/value for a follow-up test in VL.

Methods:

3. As the paper is mainly about ultrasound, ultrasound methodology should be described in more detail in the methods section. Reference 20 is a Book review of the original text with the same title; an original reference should be used (if needed). As splenomegaly and hepatomegaly are major findings, sonographic definitions of splenomegaly and hepatomegaly should be provided in the text (how were measurements performed, what were the cut-offs for definitions; comparable to the footnote to Table 3 with addition on how the liver was measured). Same for lymphadenopathy, how was abdominal lymphadenopathy defined (size of lymph nodes in what axis). Was there a definition of abnormal echotexture of organs?

Results:

4. 192 VL suspects were parasitologically confirmed; the remainder (201) were classified non-VL cases. As in the methods you mention “If no other diagnosis can be identified treatment for VL might be initiated based on RDT results” – and in Table 1 only 22.5% of non-VL patients had a concomitant infection and 58% had a positive RDT, please clarify on the final diagnoses of the 77.5% patients without VL and without other infections. Did any of these patients receive VL treatment as outlined in the methods? If so, case definitions and groups should be reconsidered. If not, please make this clear in the manuscript and comment on how confident you are that VL was excluded in these patients.

5. In Figure 1 adding the 174 spleen aspirate positive patients to the 23 bone marrow aspirate positive patients results in 197 VL positive patients; thanks for clarifying why this figure differs from the 192 confirmed patients in the text.

6. It would be very valuable to add more sonographic information on the splenic nodules. What was their echogenicity, number, size etc. This is particularly interesting as splenic nodules do also occur in other infectious diseases (e.g. TB) and to date it is unknown if these look identical or if there may be distinctive features.

7. Similarly, more detailed information on sonographic lymphadenopathy would be valuable (e.g. location, size, number, echogenicity).

8. Similarly, addition of more sonographic description of the abnormality of organ echotexture for spleen, liver, kidneys would be valuable.

9. Similarly, some more information on the quantity of pleural effusion would be valuable.

10. Was follow-up US only done in VL cases? If non-VL cases also received a follow-up scan it would be very interesting to see the evolution of findings in the absence of VL treatment.

11. To understand the sonographic follow-up data and related dynamics it is very important that you add data on the duration of treatment or time frame until follow-up US. Currently the reader cannot know if this is a few days, weeks or months.

12. In non VL patients – how where findings of liver and spleen lesion interpreted? What was the (most likely) etiology? Consider describing separately diagnoses (and course) of patients with liver or spleen lesions to better understand their possible etiology (in VL and non-VL patients).

Discussion

13. The discussion would benefit from a more critical discussion of the study’s methods and limitation (in line with the comments above). A further limitation that should be mentioned is that a linear probe was not used in all patients for assessment of the spleen; a higher detection rate of splenic nodules is achieved with linear probes and therefore in this cohort some may have had undetected nodules. The following reference shows the diagnostic difference between linear and convex probe for splenic nodules: 

Bernabeu-Wittel, M., J. L. Villanueva, J. Pachon, A. Alarcon, L. F. Lopez-Cortes, P. Viciana, F. Cadaval and A. Talegon (1999). "Etiology, clinical features and outcome of splenic microabscesses in HIV-infected patients with prolonged fever." Eur J Clin Microbiol Infect Dis 18(5): 324-329.

14. If indeed ultraound findings in VL are rare, the discussion could also include discussion of a potential value of TB ultrasound to detect concomitant TB in VL given the difficulty of confirming TB in resource-limited settings and in HIV-infected patients and particularly as concomitant TB is a risk factor for poor outcomes in VL/HIV co-infected patients.

Reviewer #3: I suggest this article is instructive and well written. I have appreciated the authors’ effort to improve US in the work-up of visceral leishmaniasis. I believe that the large sample size and the presence of a control group are the strengths of this article.

PLOS authors have the option to publish the peer review history of their article (what does this mean?). If published, this will include your full peer review and any attached files.

Reviewer #1: No

Reviewer #2: No

Reviewer #3: Yes: Francesco Taccari
---

## [Decision Letter · Decision Letter 1]

20 Nov 2020

Dear Dr. Mohammed,

Thank you very much for submitting your manuscript "Abdominal ultrasound in the diagnostic work-up of visceral leishmaniasis and for detection of complications of spleen aspiration" for consideration at PLOS Neglected Tropical Diseases. As with all papers reviewed by the journal, your manuscript was reviewed by members of the editorial board and by several independent reviewers. The reviewers appreciated the attention to an important topic. Based on the reviews, we are likely to accept this manuscript for publication, providing that you modify the manuscript according to the review recommendations. 

Please address the remining minor issues indicated by the Reviewers.

Furthermore, as suggested by the Reviewer, we strongly suggest to submit a figure/panel figure of ultrasound example images of the findings you discuss.

Sincerely,

Francesca Tamarozzi

Deputy Editor

Jaap van Hellemond

Deputy Editor

Reviewer's Responses to Questions

**Key Review Criteria Required for Acceptance?**

**Methods**

-Are the objectives of the study clearly articulated with a clear testable hypothesis stated?

-Is the study design appropriate to address the stated objectives?

-Is the population clearly described and appropriate for the hypothesis being tested?

-Is the sample size sufficient to ensure adequate power to address the hypothesis being tested?

-Were correct statistical analysis used to support conclusions?

-Are there concerns about ethical or regulatory requirements being met?

Reviewer #1: All suggested comments were taken care of.

Reviewer #2: (No Response)

Reviewer #3: (No Response)

**Results**

-Does the analysis presented match the analysis plan?

-Are the results clearly and completely presented?

-Are the figures (Tables, Images) of sufficient quality for clarity?

Reviewer #1: (No Response)

Reviewer #2: (No Response)

Reviewer #3: (No Response)

**Conclusions**

-Are the conclusions supported by the data presented?

-Are the limitations of analysis clearly described?

-Do the authors discuss how these data can be helpful to advance our understanding of the topic under study?

-Is public health relevance addressed?

Reviewer #1: (No Response)

Reviewer #2: (No Response)

Reviewer #3: (No Response)

**Editorial and Data Presentation Modifications?**

Reviewer #1: (No Response)

Reviewer #2: (No Response)

Reviewer #3: Table 2, row "Positive rK 39 RDT", column "non-VL": correct (67.7) not (677)

**Summary and General Comments**

Reviewer #1: (No Response)

Reviewer #2: The authors have very much improved their manuscript by the revisions suggested by the reviewers and I believe this is an important work for the concerned readers.

I have two remaining points, both referring to the spenic nodules seen by ultrasound - which constitute the most interesting sonographic findings in my view.

I had pointed to the importance that linear probes are used to detect splenic nodules (at least this is the case for splenic nodules due to TB). The authors replied that fortunately a linear probe was used, but in the methods it reads that a linear probe was only used to confirm e.g. splenic nodules. I therefore assume a linear probe was only used if splenic nodules were seen by convex probes. Therefore my first point remains true - if you don't use a linear probe for screening you will likely miss nodules in the spleen. It think this should be discussed (at least in the limitation sections) because reported prevalence of splenic lesions might have been an underestimate of the true prevalence.

I am surprised to read that all splenic nodesl were hyperechoic because the literature mostly reports hypoechoic splenic lesions in VL. It would be very interesting if you could provide a characteristic ultrasound image of splenic nodules in a VL patient to accompany your article

Reviewer #3: I think that the authors properly satisfied all the reviewers' questions and comments.

PLOS authors have the option to publish the peer review history of their article (what does this mean?). If published, this will include your full peer review and any attached files.

Reviewer #1: No

Reviewer #2: No

Reviewer #3: No
---

## [Decision Letter · Decision Letter 2]

5 Jan 2021

Dear Dr. Mohammed,

We are pleased to inform you that your manuscript 'Abdominal ultrasound in the diagnostic work-up of visceral leishmaniasis and for detection of complications of spleen aspiration' has been provisionally accepted for publication in PLOS Neglected Tropical Diseases.

Best regards,

Francesca Tamarozzi

Deputy Editor

Jaap van Hellemond

Deputy Editor

---

## [Editor Report · Acceptance letter]

10 Feb 2021

Dear Dr. Mohammed,

We are delighted to inform you that your manuscript, "Abdominal ultrasound in the diagnostic work-up of visceral leishmaniasis and for detection of complications of spleen aspiration," has been formally accepted for publication in PLOS Neglected Tropical Diseases.

Best regards,

Shaden Kamhawi

co-Editor-in-Chief

Paul Brindley

co-Editor-in-Chief
